# Role of Renin–Angiotensin System and Macrophages in Breast Cancer Microenvironment

**DOI:** 10.3390/diseases13070216

**Published:** 2025-07-10

**Authors:** Abir Abdullah Alamro, Moudhi Abdullah Almutlaq, Amani Ahmed Alghamdi, Atekah Hazzaa Alshammari, Eman Alshehri, Saba Abdi

**Affiliations:** Department of Biochemistry, College of Science, King Saud University, P.O. Box 2455, Riyadh 11451, Saudi Arabia; aalamro@ksu.edu.sa (A.A.A.); moudhi.a.almutlaq@gmail.com (M.A.A.); aalghamedi@ksu.edu.sa (A.A.A.); ahalshammari@ksu.edu.sa (A.H.A.); emalshehri@ksu.edu.sa (E.A.)

**Keywords:** angiotensin II (Ang II), tumor associated macrophages (TAM), breast cancer, THP-1-like macrophages, MCF-7, MDA-MB-23

## Abstract

Background/Objectives: The renin–angiotensin system (RAS) is well-established as a moderator of cardiovascular equilibrium and blood pressure. Nevertheless, growing evidence indicates that angiotensin II (Ang II), the principal RAS effector peptide, together with additional constituents, is involved in various malignancies. Since the immune system is an important aspect in tumor development, this study sought to investigate the role of Ang II in the crosstalk between tumor-associated macrophages (TAMs) and breast cancer cells in the tumor microenvironment (TME). Methods: We treated THP-1-like macrophages with 100 nM Ang II for 24 h. The culture media thus obtained was used as conditioned media and applied at 50% on MCF-7 and MDA-MB-231 breast cancer cell lines. The effects of the conditioned media on cancer cell lines were then investigated by various methods such as a cell proliferation assay, migration assay, polarization assay, and by the detection of apoptosis and reactive oxygen species (ROS) generation. Results: We demonstrated that in vitro Ang II promotes macrophage polarization toward proinflammatory M1-like macrophages and anti-inflammatory M2-like macrophages. Interestingly, Ang II, through macrophages, showed varied effects on different breast cancer cell lines, promoting tumor growth and progression in MCF-7 while inhibiting tumor growth and progression in MDA-MB-23. Conclusions: This study has provided clear evidence that Ang II in the TME modulates TAM polarization and secretions, leading to different effects based on the type of breast cancer.

## 1. Introduction

Breast cancer remains one of the most common and main causes of malignancy-related deaths among women. The disease exhibits unique epidemiological patterns and extreme heterogeneity [1]. With the escalation in the worldwide incidence of breast cancer [2], it is crucial to investigate its multifaceted nature to develop effective treatment and management strategies. The renin–angiotensin system (RAS) is an endocrine system classically implicated in the systematic regulation of hydromineral balance and renal control of blood pressure. However, besides being expressed systemically in the liver, kidney, and lung, it is also expressed locally in different tissues—such as breast, pancreas, brain, ovaries, adipose, and heart tissue. This locally expressed RAS is associated with several physiological and pathological functions including tissue remodeling, endothelial dysfunction, inflammation, and cellular proliferation [3,4,5,6]. Therefore, the imbalance of the local RAS could be a contributing factor for cancer metastasis, adhesion, invasion, angiogenesis, and proliferation. The precise role of each component of the RAS is, however, contradictory depending on the type and stage of cancer [7,8]. A review of the literature indicates that since RAS components are locally expressed in breast tissue, they may play a role in breast cancer pathology [9,10,11]. Angiotensin II (Ang II), a product of the angiotensin-converting enzyme (ACE), is the major effector peptide of the RAS. It is well recognized that Ang II acts as a growth factor, regulating cell growth and fibrosis, besides being a physiological mediator restoring circulatory integrity [9]. Studies suggest that the altered expression of the ACE has been linked to the initiation and progression of various cancers. While most of these studies conclude that overexpression of the ACE is related to poor prognosis of cancer [12,13], others indicate that the ACE may have anti-cancer effects [14,15].

For their progression and metastasis, cancer cells depend upon a complex and controlled interaction between the tumor and the anti-tumor immune response, which is facilitated by the tumor microenvironment (TME) [16]. The TME is a composite of non-cellular components including the extracellular matrix (ECM) and non-malignant stromal cells such as cancer-associated fibroblasts (CAFs) and immune cells. CAFs have been shown to liberate a range of soluble elements, which include chemokines and growth factors. These substances influence the malignancy-related stroma, promoting cancer growth and spread [17,18]. Among the immune cells, the TME abundantly comprises tumor-associated macrophages (TAMs). In response to various signaling molecules present in the TME, the TAM initially promotes anti-cancer immunity as M1-like macrophages but later, they could acquire tumor-promoting phenotypes that are called M2-like macrophages. These M2-like macrophages mainly create an immunosuppressive TME [19]. Thus, TAMs and CAFs are principal players that give rise to metabolic changes within the ECM associated with neoplasia. Interestingly, both these cells are characterized by Ang II receptor expressions and are associated with the likelihood of metastasis, a worse clinical outcome, and increased mortality [20].

The Ang II receptors, which ultimately mediate the signal transduction and functions of Ang II, have been pharmacologically recognized to be present as two subtypes, namely the angiotensin II type 1 receptor (AT1R) and the angiotensin II type 2 receptor AT2R [21]. Within the tissues, AT1R and AT2R have polar actions, so the eventual local impact of Ang II depends on the sum of the outcomes from the stimulation of these two receptor forms. Ang II/AT1R signaling has been documented to influence the TME via the facilitation of macrophage movement and infiltration into the primary cancer locus through the actions of signaling pathways encompassing a monocyte chemoattractant protein-1 [22]. In a range of malignant models, Ang II/AT1R cues have been shown to augment the production and infiltration of TAMs, whereas RAS suppressors demonstrate the ability to restrict the growth of neoplastic cells and the TAMs’ reaction [23]. These observations imply that the RAS is a key player in escalating tumor-associated inflammation within a microenvironment characterized by immunosuppressant activity. Excess Ang II synthesis could therefore represent a potential pathogenetic pathway for tumor development [9]. The local RAS could induce the accumulation of TAMs in the TME. Concurrently, several studies have revealed that Ang II plays a pivotal role during breast cancer development. It was found that the inhibition of Ang II production in the breast cancer microenvironment reduced TAM accumulation [24,25,26].

The role of AT2R activity remains a subject of debate, although a higher proportion of studies favor the concept that AT2R is associated with anti-inflammatory and tumor suppression [24,27] activities. Thus, one could postulate that AT2R functions may be repressed within the TME, highlighting possible therapeutic avenues [28,29].

The interactions between the immune system and breast cancer, though well studied, are complex, and there are significant gaps in understanding the same. Since extensive epidemiological data related to the effect of RAS inhibitors on cancer incidence and survival outcomes is inconsistent, additional mechanistic studies are warranted to investigate the precise effects of the RAS on breast cancer cells and its microenvironment [30]. Investigating the role of angiotensin II (Ang II) in modulating the interactions between tumor-associated macrophages (TAMs) and breast cancer cells within the tumor microenvironment holds significant translational relevance. The goal is to leverage this relationship for the therapeutic benefit of patients. A deeper understanding of the precise role of each individual RAS component in the TME may uncover novel molecular targets for therapeutic intervention aimed at disrupting pro-tumor immune interactions. Furthermore, given the widespread clinical use of ACE inhibitors, this research may also reveal unintended effects of modulating the renin–angiotensin system on tumor progression, thereby informing safer and more personalized treatment strategies in breast cancer patients [16,31]. Thus, our present study sought to investigate whether Ang II regulates the crosstalk between macrophages and tumoral breast cells, both noninvasive (MCF-7) and aggressive (MDA-MB-231) in the TME. Interestingly, this study provided novel evidence which suggests that Ang II, through macrophages, shows varied effects on breast cancer cell lines.

## 2. Materials and Methods

### 2.1. Cell Culture and Treatment

THP-1 human monocytic cell line (ATCC^®^ TIB-202™, Manassas, VA, USA) was cultured in Roswell Park Memorial Institute (RPMI) 1640 media supplemented with 10% fetal bovine serum (FBS) and 1% penicillin–streptomycin. Human breast cancer cell lines MCF-7 (ATCC^®^HTB-22^TM^) and MDA-MB-231(ATCC^®^HTB-26^TM^) were maintained in Dulbecco’s Modified Eagle’s minimal essential Medium (DMEM) supplemented with 10% FBS and 1% penicillin-streptomycin. All cells were maintained in a humidified 5% CO_2_ incubator (NuAire, Plymouth, MN, USA) at 37 °C. At 90% confluency, the culture media were removed and cells were washed with phosphate-buffered saline (PBS). Then, cells were detached by adding 2–5 mL TrypLE™ Express to the flask or plate for 3–5 min depending on the cell type. The cells were detached and centrifuged at 1500 rpm for 4 min at RT. The supernatant was discarded, and the cell pellet was used for further assays. All reagents used were purchased from Gibco-Invitrogen, Waltham, MA, USA.

### 2.2. Cell Counting and Viability

To determine the required volume of cells to be seeded at a specific cell density, the Vi-CELL™ XR 2.03 cell viability analyzer (Beckman Coulter, Indianapolis, IN, USA) was used. The cell suspension (500 µL) was mixed with Trypan Blue dye in the sample tube and inserted in the cell analyzer to assess the number of viable cells by excluding the dead cells which absorb the dye.

### 2.3. THP-1 Differentiation and Conditioned Media Collection

Cultured THP-1 monocyte cells were stimulated with 100 nM phorbol 12-myristate 13-acetate (PMA) (SIGMA-ALDRICH, Hamburg, Germany) for 72 h to induce monocyte differentiation into THP-1-like macrophages. Images of the differentiated cells were taken at 20× magnification using an Eclipse Ts2 inverted microscope (Nikon, Tokyo, Japan). When at least 20% of THP-1-like macrophages were in the stimulated flask, they were washed and treated with/without 100 nM Ang II (MilliporeCalbiochem, San Diego, CA, USA, Cat no. 05-23-0101-25MG) for 24 h. After that, culture media which contained Ang II-treated THP-1-like macrophages secretions was collected as an Ang II-conditioned media (Ang II-CM). Using 50% Ang II-CM diluted in DMEM, a study treatment to treat confluent MCF-7 and MDA-MB-231 cell lines for further assays was achieved. The secretions from THP-1-like macrophages untreated with Ang II were diluted with 100% completed DMEM and were considered as the study control. Thus, the study control contained PMA and the secretions from untreated THP-1-like macrophages, while Ang II-CM contained PMA, Ang II, and the secretions from AngII-treated THP-1-like macrophages. The remaining THP-1-like macrophages were washed with 1xPBS and trypsinized with TrypLE™ Express to detach adherent cells. The cells were centrifuged at 1500 rpm for 5 min at RT and the cell pellet was used for the polarization assay.

### 2.4. Reactive Oxygen Species (ROS) Generation Assay

The production of ROS by the cells was measured using the CellROX kit (Fisher Scientific, Waltham, MA, USA, Cat no. C10422). MCF-7 and MDA-MB-231 cell lines were seeded in 48-well plates at 75 × 103 and 50 × 103 cell/well, respectively, using DMEM media. After 24 h, culture media was removed and cells were treated with/without 50% Ang II-CM for 24 h and 48 h. At the end of the incubation, cells were trypsinized and stained with CellROX deep red reagent for 15 min in the dark at RT. Then, cells were washed with 1X PBS and centrifuged at 1500 rpm for 4 min. Thereafter, the cell pellet was dissolved in PBS, and a BD FACSCanto-II flow cytometer (BD Biosciences, Franklin Lakes, NJ, USA) was used for analysis.

### 2.5. Apoptosis Assay

To determine apoptosis in cells, a double-stain apoptosis detection kit (Hoechst 33342/PI) (Invitrogen, Carlsbad, CA, USA, Cat no. H3570) was used. Both MCF-7 and MDA-MB-231 cell lines were seeded in 24-well plates at 100 × 103 and 75 × 103 cell/well, respectively, using complete DMEM media. After 24 h, culture media was removed and cells were treated with/without 50% Ang II-CM for 24 h and 48 h. At the end of incubation, treated cells were harvested and washed in cold PBS, the supernatant was discarded, and cells were resuspended in 100 μL of 1× Hoechst 33342 buffer supplemented with 0.2 μL Hoechst 33342 dye and 2 μL of 100 μg/mL propidium iodide (PI) for 15 min at 37 °C. Stained cells were analyzed by a flow cytometer (emission: 486 nm and 615 nm of Hoechst 33342 and PI, respectively).

### 2.6. Cell Proliferation Assay

MCF-7 and MDA-MB-231 cells were seeded in 96-well plates at 40 × 103 and 25 × 103 cell/well, respectively, using complete DMEM media. After 24 h, culture media was removed and cells were treated with/without 50% Ang II-CM for 24 h, 48 h, and 72 h. At the end of the incubation, culture media was replaced with 100 µL phenol red free DMEM media (Gibco-Invitrogen, Waltham, MA, USA) and cells were incubated with 5 mg/mL 3-(4,5-dimethylthiazol-2-yl) 2,5-diphenyl tetrazolium bromide (MTT) reagent (Sigma, Hamburg, NY, USA) in a 5% CO_2_ incubator for 2–3 h until purple formazan crystals were formed. Next, the media was removed and 100 µL DMSO was added to dissolve formazan crystals. A SpectraMax M5 microplate reader (Molecular Devices, San Jose, CA, USA) was used to read the absorbance at 560 nm. Cell viability% was measured by using the following formula: (%) = [100 × (sample absorbance)/(control absorbance)]. 

### 2.7. Migration Assay

MCF-7 and MDA-MB-231 were seeded in 96-well plates using DMEM media. At 90% confluency, culture media was removed and the monolayer of cells was scratched using a 10 µL pipette tip. The cells were gently washed twice with 1× PBS to remove floating cells and cells were treated with/without 50% Ang II-CM. To track the wound-healing process, we used time-lapse imaging microscopy (EVOS^®^ FL Auto Imaging System, Thermo Fisher scientific, Waltham, MA, USA), where an image was taken every 4 h for 24 h. Images were analyzed using ImageJ software (version 1.54h) (W.S. Rasband, National Institutes of Health, Bethesda, MD, USA). Cells were maintained in a 5% CO_2_ chamber at 37 °C throughout the experiment.

### 2.8. THP-1-LIKE Macrophags Polarization Assay

A total of 100 nM of Ang II-treated THP-1-like macrophages were divided into two groups. The M1 group was stimulated with 50 ng/mL lipopolysaccharide and 0.05 µg/mL IFN-γ to induce cell polarization into the M1-like macrophage phenotype. The M2 group was stimulated with 20 ng/mL IL-4 to induce cell polarization into the M2-like macrophage phenotype. The cells were seeded in 6-well plates for 24 h. Next, the cells were collected and centrifuged at 1500 rpm for 4 min. Cell pellets were dissolved in 100 µL of 0.3% bovine serum albumin (SIGMA-ALDRICH, Hamburg, Germany) in PBS to prevent the non-specific binding of antibodies for 20 min at RT. The cell surface markers were stained for 1 h at 4 °C in the dark. After that, only the M1 group was permeabilized with 20 µL of 0.1% tween (BIO-RAD, Watford, UK) for 20 min at RT, and TNF-α intracellular markers were stained for 1 h at 4 °C in the dark. Finally, cells were centrifuged at 1500 rpm for 5 min at RT and the pellet was dissolved in 300 µL 1× PBS. A total of 100 nM of PMA-treated THP-1-like macrophages was considered as a control. All antibodies used for staining and their sources are listed in Table 1. A BD FACSCanto-II flow cytometer (BD Biosciences, Franklin Lakes, NJ, USA) and BD FACSDiva software were used for cell measurement and analysis.

### 2.9. Enzyme-Linked Immunosorbent Assay

IL-10 (SEKH-0018, Solarbio Life Sciences, Beijing, China) and IL-17 (BMS2037-2, Invitrogen, Carlsbad, CA, USA) ELISA kits were used to determine the concentration of these anti-inflammatory cytokines. All reagents and standards were prepared as directed by the manufacturers. A total of 100 µL of standards or samples were added to each well and incubated for 90 min at 37 °C. The plate was washed 4 times with washing buffer. Then, 100 µL of the biotin-conjugated detection antibody was added to each well and incubated for 60 min at 37 °C. The plate was washed 4 times with washing buffer. In total, 100 µL of streptavidin–HRP was added to each well and incubated for 30 min at 37 °C. Then, the plate was washed 5 times with the washing buffer. After that, 100 µL of substrate solution (TMB) was added to each well and incubated for 15 min at 37 °C, protected from light. Finally, 50 µL of stop solution was added to each well and the plate was read at 450 nm within 30 min.

### 2.10. Statistical Analysis

Each experiment was performed in triplicate. Results were represented as mean ± SD from at least three independent experiments. Two-tailed *t*-tests were used for data analysis. All *p*-values < 0.05 were considered statistically significant.

## 3. Results

### 3.1. THP-1 Differentiation

The THP-1 monocytic cells were stimulated with PMA to induce monocyte differentiation into THP-1-like macrophages. Light microscopy analysis revealed that changes in PMA induced morphology, including increased cellular adhesion and spread morphology. Thus, PMA-stimulated THP-1 monocytic cells showed the THP-1-like macrophage phenotype (Figure 1a,b).

### 3.2. ROS Generation Assay

The level of ROS is a direct indicator of oxidative stress. CellROX deep red reagent is widely used to measure the level of ROS in living cells. It contains non-fluorescent dye in a reduced state, which is oxidized by free radicals present in the living cells and becomes fluorescent. In the present study, we examined whether Ang II has the potential to evoke oxidative stress in breast cancer cells. In MCF-7 treated with 50% Ang II-CM, there was a significant (*p* < 0.05 vs. control) decrease in ROS levels after 24 h of treatment (Figure 2a). In contrast, the level of ROS in MDA-MB-231 treated with 50% Ang II-CM was significantly increased after 48 h (*p* < 0.01 vs. control) (Figure 2b).

### 3.3. Apoptosis Assay

An apoptosis assay was performed using Hoechst 33342 /PI staining to investigate the type of cell death undergone by each treated cell line. MCF-7 treated with 50% of Ang II-CM showed a significant change in the ratio of dead, apoptotic, and necrotic cells. There was a decrease (*p* < 0.05 vs. control) in the ratio of dead cells after 24 h of treatment (Figure 3a). Conversely, the ratio of apoptotic cells (Figure 3b) demonstrated a significant decrease (*p* < 0.05 vs. control) and the necrotic cell ratio (Figure 3b) exhibited a significant increase (*p* < 0.01 vs. control) after 48 h of treatment compared to the control. In contrast, 50% Ang II-CM-treated MDA-MB-231 cells showed a different behavior compared to MCF-7 cells, in which the ratio of apoptotic cells and dead cells was highly increased but the ratio of necrotic cells decreased at both time points of treatment, although the observed results were non-significant (Figure 3c,d).

### 3.4. Proliferation Assay

MTT reagent was used to assess the percentage of viable cells in response to 50% of Ang II-CM treatment. As shown in Figure 4a, when compared to the controls, MCF-7 did not show any significant change in the percentage of viable cells following treatment. However, MDA-MB-231 showed a significant decrease in the number of viable cells compared to the control when treated for 24 h, 48 h, and 72 h (*p* < 0.01, *p* < 0.05, *p* < 0.05, respectively, vs. control) (Figure 4b).

### 3.5. Cell Migration Assay

For the cell migration assay, 50% of Ang II-CM was applied on the cell-free area after making a scratch on the cell monolayer of MCF-7 and MDA-MB-231. The wound-healing process was tracked using the EVOS imaging system within a 24 h time frame. The rate of the wound-healing process in MCF-7 cells was three times faster than the control group, as shown in Figure 5 and Figure 6. Intriguingly, the wound-healing process in the MDA-MB-231 cell monolayers was not proper. Instead, the morphology of MDA-MB-231 cells changed from a spindle to a spherical shape, and cells started to aggregate (Figure 7). Moreover, it was not possible to measure the wound closure area and perform a time course analysis for MDA-MB-231 cells.

### 3.6. THP-1-Like Macrophage Polarization Assay

The polarization assay was performed to investigate the effect of 100 nM Ang II on macrophage polarization. CD14 is a macrophage-specific differentiation antigen [32]. It was used to identify differentiated THP-1-like macrophages.

The relative expression of cell surface markers such as TNF-α, MHC-II, and HLA-DR was used to examine the polarization of macrophages to the M1 phenotype, while CD206 was used as a marker of macrophage polarization toward the M2 phenotype. As shown in Figure 8, there was an increase in differentiated THP-1-like macrophages after Ang II treatment. Moreover, significant increases in TNF-α- and MHCII-expressing cells were detected, indicating the M1 phenotype, while the detection of significant increases in CD206-expressing cells indicated M2 phenotypes.

### 3.7. Enzyme-Linked Immunosorbent Assay

To estimate cytokine secretion, an ELISA was performed using conditioned media from THP-1-like macrophages treated with 100 nM Ang II (Figure 9). Increased IL-10 and IL-17 cytokines were detected in the conditioned media, which is higher than in the control groups by 1.3-fold and 2.33-fold, respectively (Figure 10).

## 4. Discussion

The local RAS, present in different human tissues, may play a role in local tissue disorders [28,33]. Its dysregulation has been observed during the malignant transformation of different tissues as well as during cancer development and progression. For example, breast cancer expression of the RAS component AT1R has been found to increase during the ductal carcinoma stage, while it decreases during the invasive carcinoma stage [34]. AT1R polymorphisms have been discovered in ethnic populations, and specific polymorphic genotypes have been linked to a predisposition to develop more aggressive breast cancer [35]. Similarly, ACE gene polymorphisms has been known to cause RAS overactivation [36].

Interestingly, the effect of the local RAS is not only confined to cancer cells but extends to affect the cellular and molecular components in its environment. TAMs, an important part of the tumor environment, are well known to express RAS components. They promote cancer development and progression through modulating the environment [30]. Despite extensive research on both the RAS and TAMs in cancer biology, few in vitro studies have directly addressed the involvement of Ang II in the crosstalk between TAMs and breast cancer cells in the tumor microenvironment. Our current study focuses on AngII, a major effector component of the RAS. We hypothesized that Ang II might regulate the interaction between macrophages and cancer cells in the breast cancer microenvironment. To test this hypothesis, Ang II was used to treat THP-1-like macrophages for 24 h, then culture media was used as the conditioned media and applied at 50% on MCF-7 and MDA-MB-231 breast cancer cell lines at different time points to investigate how Ang II affects the interaction between TAMs and the breast cancer microenvironment.

Macrophages are present abundantly in the tumor milieu, where in response to various surrounding signaling molecules, they initially promote anti-cancer immunity as M1-like macrophages but later, they can acquire tumor-promoting phenotypes that are called M2-like macrophages [37]. M1-like macrophages are characterized by HLA-DR, MHC-II, and TNF-α expression, whereas M2-like macrophages can be identified by elevated expressions of CD206, CD204, and CD163 [38].

A previous study demonstrated that Ang II promotes the polarization of THP-1-derived macrophages to the M1-like macrophage phenotype [39], as evident from the enhanced expression of proinflammatory markers (HLA-DR, TNF-α, CD64, CD11c, and CD38). Our current research agrees with this study because a marked increase in TNF-α- and MHCII-expressing cells was detected. However, our research was distinctive, as we also detected a significant increase in CD206 (M2-like macrophage marker)-expressing macrophages compared to the control group.

Moreover, when we explored the contents of Ang II-CM for cytokines, it was revealed that IL-10 and IL-17 (anti-inflammatory cytokines) concentrations were increased compared to the control. IL-10 is one of the cytokines that can induce the polarization of macrophages to the M2-like phenotype. Similarly, IL-17 is involved in contributing to M2 macrophage differentiation via NF-κB activation [40]. Since M2-like macrophages support tumor growth, this can explain the growth observed in MCF 7 in our current study.

An increased production of anti-inflammatory IL-10 and IL-17 by M2-like macrophages has been found to induce the expression of ECM degradation enzymes and angiogenic markers, thus supporting tumor angiogenesis and migration [41]. Moreover, the upregulation of IL-10 expression in TAMs has been reported to contribute to lung cancer immunosuppression and cancer progression [41]. The TAMs infiltrating triple-negative breast cancer cells secrete IL-10, which contributes to tumor progression [42]. Thus, the increased IL-10 and IL-17 production by Ang II-treated THP-1-like macrophages, as our results demonstrate, perhaps plays a role in increased MCF-7 migration ability after the application of 50% Ang II-CM.

Ang II activity is well known to induce ROS production in cardiovascular diseases. ROS has been shown to stimulate both the M1 and M2 activation states of macrophages. While some studies suggest that ROS production promotes M2 polarization [43], others demonstrate that it promotes M1 polarization [44,45]. In tumor cells, ROS production results in oxidative stress, which eventually induces cell apoptosis, cell necrosis, and proinflammatory signals [28,46]. Excessive ROS generation increases mitochondrial membrane permeability that allows the inner mitochondrial membrane-bounded cytochrome c to be released in the cytoplasm and form the apoptosome complex which drives the apoptosis process [46]. In our study, ER-negative MDA-MB-231 showed a significant increase in the level of ROS, cell apoptosis, and death following 24 and 48 h of treatment, which might indicate that Ang II, through macrophages, induced ROS production, which promotes cell apoptosis and death. Apoptosis is a normal process in which defected and infected cells are removed. However, cancerous cells evade this process to proliferate and survive [47]. In conditions of low tissue oxygen levels and inflammation, Ang II can stimulate macrophages to generate ROS. This additionally amplifies RAS signaling, resulting in enhanced proliferation and angiogenesis [28]. In prostate cancer, Ang II/AT1R signaling was found to induce ROS production, which promotes angiogenesis and inflammatory signals [48].

In contrast to the result we obtained for ER-negative MDA-MB-231, our study on ER-positive MCF-7 showed a significant decrease in the level of ROS, cell apoptosis, and death after 24 and 48 h of 50% Ang II-CM treatment. This observation is consistent with a study conducted by Zhao et al., which revealed that the adriamycin-induced apoptosis of MCF-7 was attenuated after Ang II treatment [49]. The MCF-7 cell line is estrogen receptor (ER)-positive and HER2-negative; thus, it demonstrates an over-expression of AT1R.

Ang II, through macrophages, shows different effects on breast cancer cell lines, promoting tumor growth and progression in MCF-7. AT1R is profoundly upregulated in protumoral macrophages, and it causes the transactivation of the epidermal growth factor receptor (EGFR) signaling pathway. This leads to the secretion of matrix metalloproteinases (MMPs) that activate the PI3K/AKT signaling cascade [50]. Thus, RAS signaling activation through AT1R fosters the activation of transcription factors (NF-κB, STAT 3) that induce various tumor growth and progression factors (IL-6, IL-8, MCP-1, macrophage colony-stimulating factor (M-CSF), VEGF, tissue inhibitor of metalloproteinase 1 (TIMP1)) integral to the TME [51].

Besides the effects produced by Ang II via AT1R, some effects of Ang II on macrophages are dependent on AT2R stimulation [39]. AT2R is associated with anti-inflammatory activities. Cell necrosis is accidental cell death caused by physical or chemical stress from heat, radiation, oxidative stress, inflammation, etc. It is characterized by cell swelling and plasma membrane rupture, leading to the release of cell content in the surrounding environment, which in turn worsens inflammation and promotes cancer development and progression [47,52]. The present study exhibited that, compared to the control group, there was an increase in cell necrosis and proliferation in MCF-7 after 24, 48, and 72 h of applying 50% Ang II-CM, whereas an opposite effect following the same treatment was observed in MDA-MB-231. In this context, Ang II-treated macrophages showed an increase in TNF-α expression, which might have caused the cell necrosis observed in the MCF-7 breast cancer cell line. TNF-α plays a role in promoting cell necrosis, which in turn reinforces cancer progression by releasing cell content and debris [53].

Metastasis is a critical step for cancer progression. Intercellular matrix degradation, epithelial–mesenchymal transitions (EMTs), and new blood vessel formation are the main characteristics of cancer migration. Ang II/AT1R activity has previously been shown to promote the invasiveness of ovarian cancer [28]. A wound-healing assay was performed by us to assess the effect of 50% Ang II-CM on the migration ability of breast cancer cell lines during 24 h of treatment. The time course analysis showed a significant increase in MCF-7 migration ability, which was three times faster than observed in the control group. In MDA-MB-231, the migration ability was not measurable, but there was a clear change in the MDA-MB-231 morphology from a mesenchymal-like shape (spindle) to an epithelial-like shape (rounded). Furthermore, cells had started to aggregate, indicating a reverse EMT that inhibits cell migration and induces cell death. Additionally, cancer cell apoptosis, which is an important process for preventing cancer migration [54], was observed in the MDA-MB-231 breast cancer cell line.

All these findings suggest that Ang II, through macrophages, showed a differential effect on MCF-7 and MDA-MB-231 breast cancer cell lines. In ER-positive MCF-7, it promoted tumor growth and migration by decreasing ROS generation and cell apoptosis and increasing cell proliferation and migration. In ER-negative MDA-MB-231, it promoted tumor suppression by increasing ROS generation and cell apoptosis and decreasing cell proliferation and migration.

The major limitation of this study was that it did not scan whole compositions of Ang II-conditioned media, which could have provided more information on the effect of Ang II on the TAMs and breast cancer microenvironment. Future studies using animal models to investigate the role of the RAS in the crosstalk between TAMs and cancer cells could provide further insights into the precise role of each individual RAS component in the TME and help to identify possible therapeutic targets for cancer treatment.

## 5. Conclusions

The study results show the ability of Ang II to induce both the M1- and M2-like macrophage phenotypes, which in turn could promote or suppress tumor activity based on the type of cancer cell. Ang II, through macrophages, showed an increase in tumor activity in the MCF-7 breast cancer cell line. However, it suppressed tumor activity in MDA-MB-231. This indicates the complex nature of RAS signaling during tumor development that resulted in different net effects depending on the tissue type and tumor stage. Further assays should be performed to validate this observation.

## Figures and Tables

**Figure 1 diseases-13-00216-f001:**
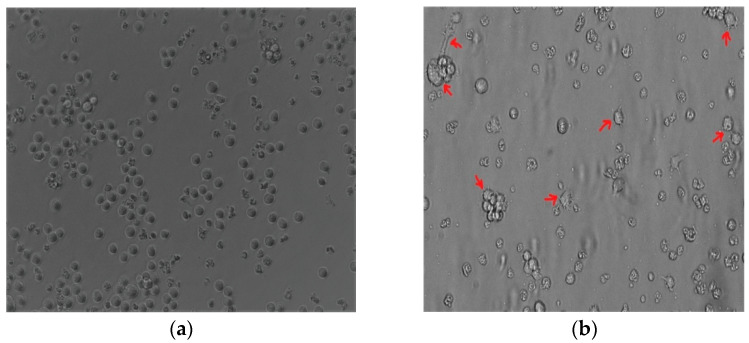
The 100 nM PMA-stimulated THP-1. (**a**) THP-1 monocyte before differentiation. (**b**) Differentiated THP-1-like macrophages. Red arrows indicate the adherent macrophage-like morphological characteristics. Images were taken at 20× magnification using an Eclipse Ts2 inverted microscope (Nikon, Tokyo, Japan).

**Figure 2 diseases-13-00216-f002:**
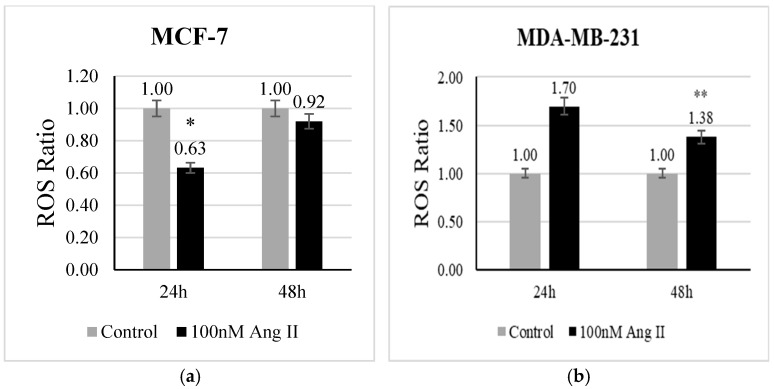
Reactive oxygen species assay. CellROX deep red reagent was used to measure ROS level in (**a**) MCF-7 treated with 50% Ang II-CM for 24 h and 48 h and (**b**) MDA-MB-231 treated with 50% Ang II-CM for 24 h and 48 h. The ROS ratio was calculated as the level of ROS in the sample to the level of ROS in the control. * Indicates *p* < 0.05 and ** indicates *p* < 0.01.

**Figure 3 diseases-13-00216-f003:**
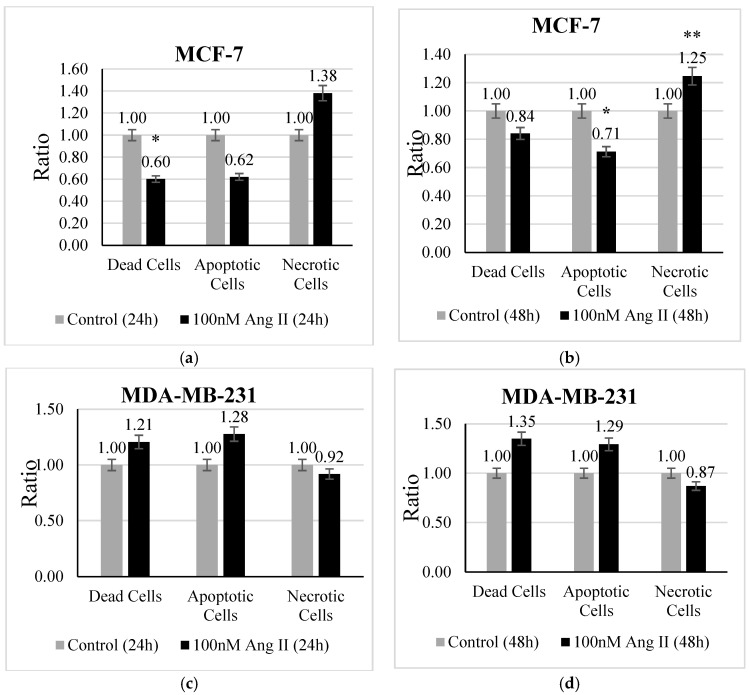
The apoptosis assay. The upper panel represents MCF-7 cells treated with 50% Ang II-CM for (**a**) 24 h and (**b**) 48 h. The lower panel represents MDA-MB-231 cells treated with 50% Ang II-CM for (**c**) 24 h and (**d**) 48 h. A flow cytometer was used to differentiate the type of cell death based on fluorescence intensity. * Indicates *p* < 0.05 and ** indicates *p* < 0.01.

**Figure 4 diseases-13-00216-f004:**
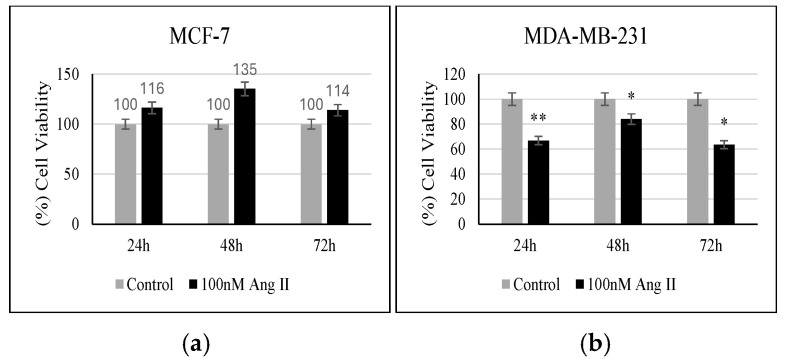
Proliferation assay. (**a**) Percentage (%) of MCF-7 viability after 24, 48, and 72 h of treatment. (**b**) Percentage (%) of MDA-MB-231 viability after 24, 48, and 72 h of treatment. * Indicates *p* < 0.05 and ** indicates *p* < 0.01.

**Figure 5 diseases-13-00216-f005:**
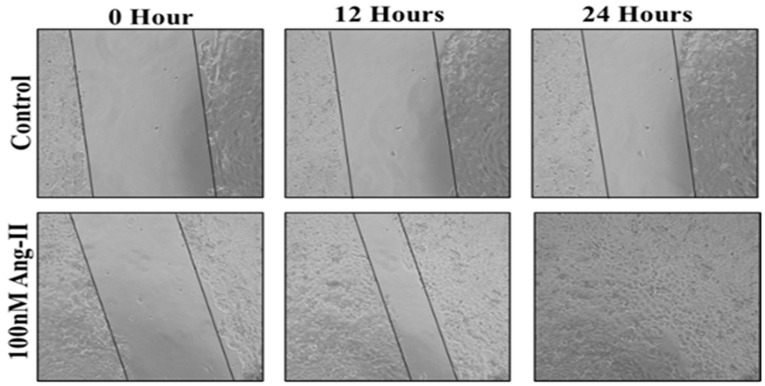
A representative image of the MCF-7 wound-healing process for 24 h of 50% Ang II-CM application. Images were taken at 10× magnification, and a time course analysis was performed using ImageJ software.

**Figure 6 diseases-13-00216-f006:**
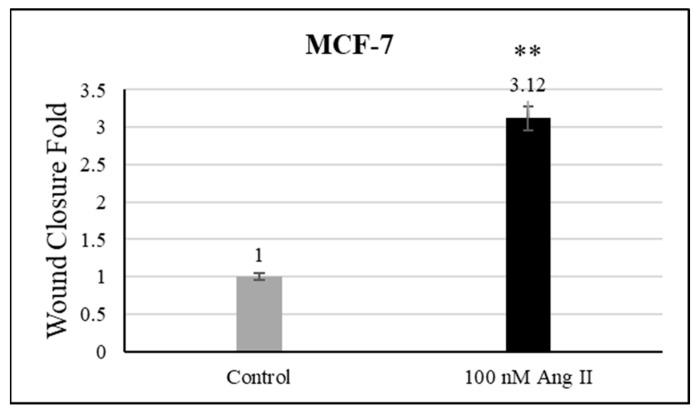
Wound closure fold of MCF-7 cultured in 50% Ang II-CM. ** indicates *p* < 0.01.

**Figure 7 diseases-13-00216-f007:**
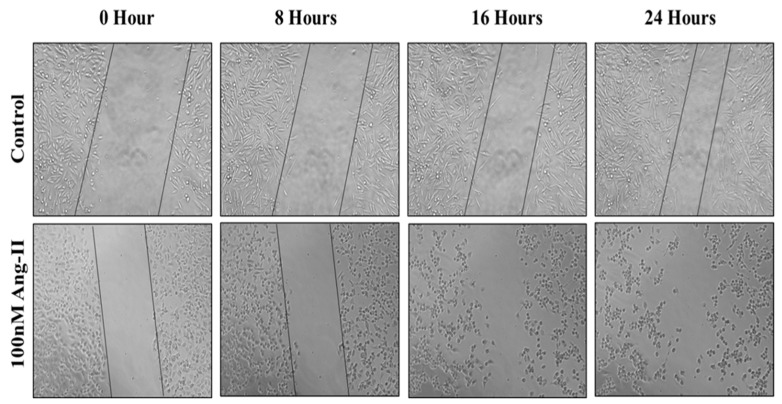
A representative image of the MDA-MB-231 wound-healing process during 24 h of 50% Ang II-CM application. Images were taken at 10× magnification, and a time course analysis was performed using ImageJ software.

**Figure 8 diseases-13-00216-f008:**
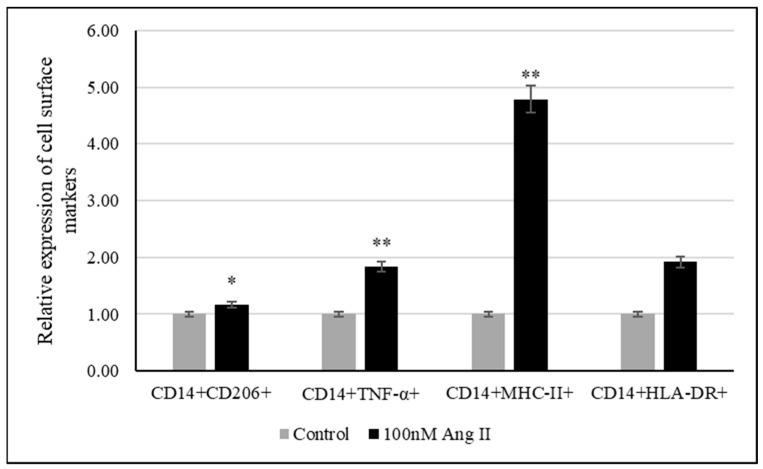
Relative expression of polarization markers after treating THP-1-like macrophages with 50% of Ang II-CM. * Indicates *p* < 0.05 and ** indicates *p* < 0.01. Flow cytometer was used to identify type of cell polarization based on fluorescence intensity of each macrophage-specific marker.

**Figure 9 diseases-13-00216-f009:**
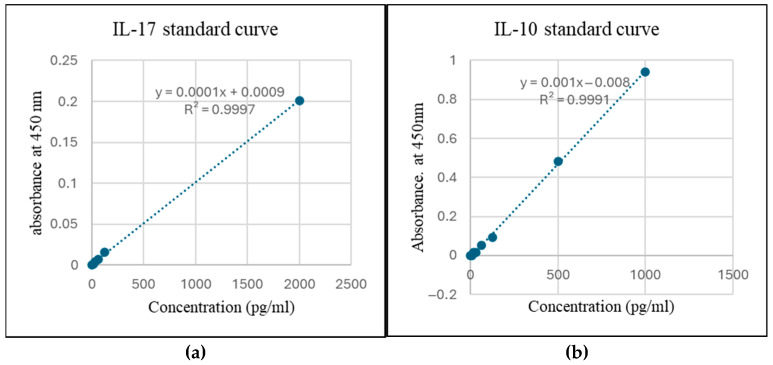
Standard curve for estimation of (**a**) IL-17 and (**b**) IL-10.

**Figure 10 diseases-13-00216-f010:**
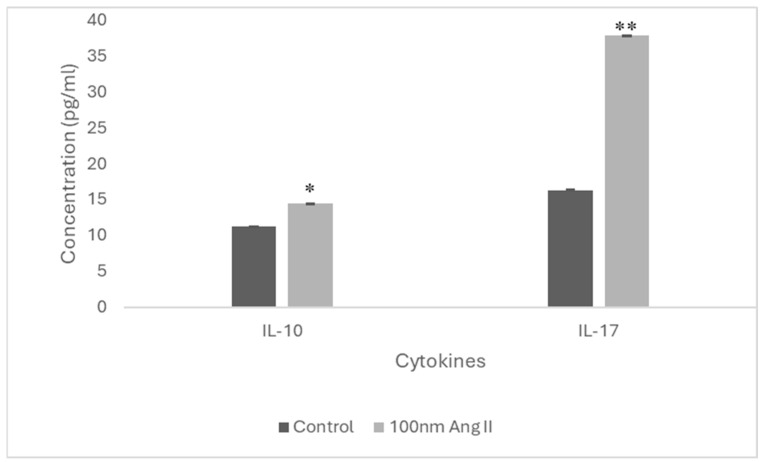
IL-17 and IL-10 concentration in Ang II-CM using ELISA. * Indicates *p* < 0.05 and ** indicates *p* < 0.01.

**Table 1 diseases-13-00216-t001:** List of antibodies used for THP-1-like macrophage polarization assay.

Antibody	Company	Catalog Number
Interleukin-4 (IL-4) human	Enzo life sciences, New York, NY, USA	ENZ-PRT180-0020
Interferon gamma (IFN-γ)	Origene technologies, Rockville, MD, USA	TP723709
Anti-M0 CD14	Invitrogen, Carlsbad, CA, USA	45-0141-80
CD206 (MMR)-PE	Beckman Coulter, Brea, CA, USA	IM2741
TNF-α-PE	Beckman Coulter, Brea, CA, USA	IM3279U
Anti-M0 MHC class II	Invitrogen, Carlsbad, CA, USA	17-5320-82
Human HLA-DR	Life technologies, Carlsbad, CA, USA	MHLDR28

## Data Availability

The data included in this article are available upon reasonable request.

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
