# Peer review of "Role of Renin–Angiotensin System and Macrophages in Breast Cancer Microenvironment"

_diseases, 2025, doi:10.3390/diseases13070216_

Round 1
Reviewer 1 Report
Comments and Suggestions for Authors
Abir Alamro and colleagues submitted an interesting investigation on the role of renin–angiotensin system and macrophages in breast cancer microenvironment. The aim of the study was to determine whether angiotensin II (ANG II) regulates the crosstalk between macrophages and tumoral breast cells, noninvasive (MCF-7) and aggressive (MDA-MB-231) in tumor microenvironment (TME). The final conclusion demonstrates that Ang II in TME modulates tumor-associated macrophages (TAMs) polarization and secretions leading to different effects based on the type of breast cancer. Overall, the manuscript is interesting; however, to improve it, the authors should address several points in the following sections, as detailed below:
General comments:
The innovative character of the paper can be seriously questioned, considering that out of 50 references, only 9 (18%) were published within the last five years, while as many as 23 (46%) date back to before 2015. The authors should incorporate more recent literature, which could strengthen the rationale and highlight the relevance of the study more clearly.
Major points:
1. It would strengthen the manuscript if the authors articulated the novelty of their work more clearly in the final paragraph of the Introduction.
2. The authors should strongly emphasize the unique aspects of their paper. How it contributes value to the field. I recommend adding a separate paragraph in the Discussion section to specifically address these points.
3. The authors should discuss the extent to which their findings can be extrapolated to human biology. A thoughtful consideration of this aspect would significantly enhance the translational value and broader impact of the work.
Minor points:
1. To enhance the clarity of the text, the authors are advised to limit the use of abbreviations in the discussion section.
Author Response
Comment 1(General comment): The innovative character of the paper can be seriously questioned, considering that out of 50 references, only 9 (18%) were published within the last five years, while as many as 23 (46%) date back to before 2015. The authors should incorporate more recent literature, which could strengthen the rationale and highlight the relevance of the study more clearly.
Response 1:The authors thank you for pointing this out and we graciously agree to this comment. We have updated the list of references. In addition to the previous 9 references ,we have now incorporated 15 new references ( can be seen in red font color in list of references )that are dated within last five years. In total more that 45% references are now dated within last 5 years.
Comment 2:It would strengthen the manuscript if the authors articulated the novelty of their work more clearly in the final paragraph of the Introduction.
Response 2: The authors are in full agreement with the suggestion provided by the reviewer. Indeed highlighting the novelty of the presented work is essential. Therefor in accordance to this comment, the final paragraph of introduction ( line 92-108, marked in red)has been revised to clearly describe the gap in knowledge regarding interactions between the immune system and breast cancer , as well as to highlight the novel findings of the study.
Comment 3:The authors should strongly emphasize the unique aspects of their paper. How it contributes value to the field. I recommend adding a separate paragraph in the Discussion section to specifically address these points.
Response 3: This comment from the reviewer is also considered valuable by the authors and we agree to it. As suggested , Line 342-346 (marked in red) in discussion section mentions the novelty of the presented work. The uniqueness of our paper lies in the fact that we have provided evidences that prove that Angiotensin II can directly drive not only M1 ( as suggested previously by several researchers) but also M2 polarization of macrophages .These evidences include enhanced expression of cell surface markers as well as cytokines specific to M2-like macrophages. This has been mentioned in Discussion section (line 358 to 370 , marked in red).
Comment 4:The authors should discuss the extent to which their findings can be extrapolated to human biology. A thoughtful consideration of this aspect would significantly enhance the translational value and broader impact of the work.
Response 4: The authors are grateful that the reviewer provided this valuable suggestion for improving the submitted manuscript. We agree to this suggestion and have revised the manuscript accordingly. The present study has direct translational relevance, as elucidating how Ang II influences TAM–breast cancer cell interactions may identify new therapeutic targets to reprogram the tumor microenvironment. It may also uncover potential adverse effects of ACE inhibitors on tumor biology, providing critical insight into the safe use of these drugs in cancer patients.This has now been mentioned in the final paragraph of introduction.(Line 92-105 , marked in red).
Comment 5: Minor points: To enhance the clarity of the text, the authors are advised to limit the use of abbreviations in the discussion section.
Response 5: We agree that repetitive use of abbreviations reduces clarity of discussion , in view of this at several places the abbreviations have been replaced by a short synonyms in discussion section.

Reviewer 2 Report
Comments and Suggestions for Authors
An interesting study in which the authors studied the role of Ang II in the interactions between tumor-associated macrophages and breast cancer cells in the tumor microenvironment. As a result of the experiments, the authors showed that Angie in TME modulates the polarization and secretion of TAMC, which leads to different effects depending on the type of breast cancer. There are the following requests for work: 1. The first paragraph of the introduction of the article should be devoted to a description of the prevalence and medical and social significance of the disease under study - breast cancer. 2. Is there any genetic data (GWAS or association studies) on the association of RAS gene polymorphism (Ang II, etc.) with breast cancer? These data should be used when discussing the results obtained. 3. Is there evidence of genetic correlations between diseases in which RAS is significantly involved and breast cancer?
Author Response
Comment 1: The first paragraph of the introduction of the article should be devoted to a description of the prevalence and medical and social significance of the disease under study - breast cancer.
Response 1: The authors thank you for pointing this out and agree to this comment. The introduction section has now been revised accordingly .Description of the prevalence and medical and social significance of the disease under study - breast cancer has now been incorporated in the 1st paragraph of introduction section ( Line 31 to 35, marked in red) .
Comment 2:Is there any genetic data (GWAS or association studies) on the association of RAS gene polymorphism (Ang II, etc.) with breast cancer? These data should be used when discussing the results obtained.
Response 2: As rightly suggested by the reviewer ,indeed there are studies on the association of RAS gene polymorphism (Ang II, etc.) with breast cancer, however , since our research is focused specifically on the role of Angiotensin II (Ang II) in the interactions between tumor-associated macrophages (TAMs) and breast cancer cells in vitro, thus discussing RAS gene polymorphisms would be outside the main scope and adding these details might make the discussion section more complex. Yet , considering the importance of this information ,the association of polymorphism in RAS components is now briefly mentioned in the first paragraph of discussion (Line 335-338, marked in red).
Comment 3:Is there evidence of genetic correlations between diseases in which RAS is significantly involved and breast cancer?
Response 3:
The authors thank the reviewer for mentioning this point. Yes, there is evidence for genetic correlations between various types of cancers and RAS components. This is evident from involvement of RAS components in tissue remodelling, endothelial dysfunction, inflammation and cellular proliferation as mentioned in line39 to 41(marked in red) of introduction section.

Reviewer 3 Report
Comments and Suggestions for Authors
This manuscript describes experiments investigating angiotensin effects on macrophages and effects of angiotensin conditioned macrophage medium on breast cancer cells. The role of angiotensin on breast cancer cells and their microenvironment is not sufficiently elucidated. The topic of this investigation is therefore relevant and fits to the scope of this journal. The manuscript is also well written.
The major result of this investigation is that angiotensin effects macrophage polarization (as observed earlier) and that conditioned medium has differential effects on breast cancer cell lines, may depending on subtype.
The authors report several relevant effect of macrophages and breast cancer cells. Interestingly, MCF-7 as luminal A cell-line and MDA-MB-231 as TNBC reacted differentially, often reciprocal to the conditioned medium.
It did not become clear, however, what the control treatments for the conditioned medium experiments are. The conditioned medium contains not only growth factors secreted by the macrophages, but also angiotensin and may be also residual PMA. It is therefore not clear whether the observed effects are due to substances released by the macrophages or remaining stimulants. Please clarify this in the method section and, if necessary, perform controls with the stimulants alone.
Apparently the wound healing assay contains serum, so the result is a combination of proliferation and migration. This should be discussed.
It would be helpful to show representative flow cytometry data leading to the data in Fig. 8.
Fig. 9 is not really necessary.
Author Response
Comment 1:
It did not become clear, however, what the control treatments for the conditioned medium experiments are. The conditioned medium contains not only growth factors secreted by the macrophages, but also angiotensin and may be also residual PMA. It is therefore not clear whether the observed effects are due to substances released by the macrophages or remaining stimulants. Please clarify this in the method section and, if necessary, perform controls with the stimulants alone.
Response1 : The authors agree that the control treatment was not clearly defined in the submitted manuscript. We thank the reviewer for pointing this out. The conditioned media (Ang II-CM) contained Ang II treated-THP-1 secretions. 50% of Ang II-CM diluted in DMEM was used as study treatment to treat confluent MCF-7 and MDA-MB-231 cell lines .While secretions from THP-1 like macrophages untreated with Ang II and treated with 100% completed DEME were used as study control to treat confluent MCF-7 and MDA-MB-231 cell lines.Thus the study control was with stimulant and the secretions from untreated THP-1 like macrophages while the conditioned media had stimulant , Ang II as well as the secretions AngII treated THP-1 like macrophages. The method section ( line 140-144, marked in red ) has now been revised to clearly mention this information.
Comment 2:
Apparently the wound healing assay contains serum, so the result is a combination of proliferation and migration. This should be discussed.
Response 2:
The authors would like to thank the reviewer for their concern , however for the cell migration assay, 50% of Ang II-CM was applied on the cell free area after making a scratch on the cell monolayer of MCF-7 and MDA-MB-231, so there is no possibility of presence of serum in the assay.
Comment 3:It would be helpful to show representative flow cytometry data leading to the data in Fig. 8.
Response 3: The authors agree to this comment and will provide the flow cytometry raw data as supplementary material is the reviewer considers it necessary.
Comment 4: Fig. 9 is not really necessary.
Response 4: The authors agree to this comment. However this figure was not removed as some other reviewers might request standard curves for ELISA.

Round 2
Reviewer 1 Report
Comments and Suggestions for Authors
The authors have improved the article according to my suggestions. I suggest to publish the article in its current form
Reviewer 3 Report
Comments and Suggestions for Authors
Thank you for submitting a revised version of your manuscript. I think my points of concern have been answered sufficiently.